

# Histological study of seventeen organs from dugong (*Dugong dugon*)

Patcharaporn Kaewmong[1], Pathompong Jongjit[1], Araya Boonkasemsanti[1], Kongkiat Kittiwattanawong[1], Piyamat Kongtueng[2,3], Pitchaya Matchimakul[4], Wasan Tangphokhanon[4], Prapawadee Pirintr[4], Jaruwan Khonmee[3,4], Songphon Buddhasiri[3,4], Promporn Piboon[3,4], Sonthaya Umsumarng[3,4], Raktham Mektrirat[4], Korakot Nganvongpanit[3,4] and Wanpitak Pongkan[3,4]

[1] Phuket Marine Biological Center, Phuket, Thailand
[2] Central Laboratory, Faculty of Veterinary Medicine, Chiang Mai University, Chiang Mai, Thailand
[3] Research Center for Veterinary Biosciences and Veterinary Public Health, Chiang Mai University, Chiang Mai, Thailand
[4] Department of Veterinary Biosciences and Veterinary Public Health, Faculty of Veterinary Medicine, Chiang Mai University, Chiang Mai, Thailand

Corresponding author
Wanpitak Pongkan,
P.wanpitak@gmail.com

## ABSTRACT

**Background:** Dugongs are marine mammals with a crescent-shaped tail fluke and a concave trailing margin that belong to the family *Dugongidae*., They are distributed widely in the warm coastal waters of the Indo-Pacific region. Importantly, the population of dugongs has decreased over the past decades as they have been classified as rare marine mammals. Previous studies have investigated the habitat and genetic diversity of dugongs. However, a comprehensive histological investigation of their tissue has not yet been conducted. This study provides unique insight into the organs of dugongs and compares them with other mammal species.

**Methods:** Tissue sections were stained with Harris's hematoxylin and eosin Y. The histological structure of 17 organ tissues obtained from eight systems was included in this study. Tissue sections were obtained from the urinary system (kidney), muscular system (striated skeletal muscle and smooth muscle), cardiovascular system (cardiac muscle (ventricle), coronary artery, and coronary vein), respiratory system (trachea and lung), gastrointestinal system (esophagus, stomach, small intestine, liver, and pancreas), reproductive system (testis), lymphatic system (spleen and thymus), and endocrine system (pancreas).

**Results:** While most structures were similar to those of other mammal species, there were some differences in the tissue sections of dugongs when compared with other mammalian species and manatees. These include the kidneys of dugongs, which were non-lobular and had a smooth, elongated exterior resulting in a long medullary crest, whereas the dugong pyloric epithelium did not have overlying stratified squamous cells and was noticably different from the Florida manatee.

**Discussion:** Histological information obtained from various organs of the dugong can serve as an essential foundation of basal data for future microanatomical studies. This information can also be used as high-value data in the diagnosis and pathogenesis of sick dugongs or those with an unknown cause of death.

# INTRODUCTION

Dugongs are marine mammals with a crescent-shaped tail fluke and a concave trailing margin that belong to the family *Dugongidae,*. They are distributed widely in the warm coastal waters of the Indo-Pacific region (*Marsh & Sobtzick, 2015*; *Plon et al., 2019*). The dugong (*Dugong dugon*) is the surviving species of the family and is critically endangered, as has been confirmed in Appendix I of the International Union for Conservation of Nature's Red List of Threatened Species (*Marsh, O'Shea & Reynolds, 2012*). This marine species is a critical component of the food chain, the biological diversity, and the long-term sustainability of circulating oceanic systems (*Lotze, 2021*). Unfortunately, dugong populations are believed to be close to extinction in the waters of Japan, Hong Kong, Taiwan, the Philippines, Cambodia, Vietnam, the Maldives, and Mauritius (*Lawler et al., 2002*). In Thailand, the population of the dugong has been reported to be less than 200 animals (*Pradip Na Thalang, Thongratsakul & Poolkhet, 2023*). This is consistent with the outcomes of previous studies, which have reported that there are very few dugongs in existence in the Gulf of Thailand (*Adulyanukosol, 2006*). The principal causes for the stranding of restricted dugongs include terrain and climate changes, ocean contamination, natural infections, tidal disturbances, anthropogenic threats, and misdirected navigation. The genetic diversity of dugongs is currently less than that of past decades (1990–2002) since habitat fragmentation on dugong populations has occurred (*Poommouang et al., 2021*). Previous studies have reported a reduction in the genetic diversity of dugong populations throughout the Western Indian Ocean (*Plon et al., 2019*). In addition, dugongs have also been classified as rare marine mammals by the Wild Animal Reservation and Protection Ac t, B.E.2553 (*Hines et al., 2005*).

In response to this decline in population, effective health monitoring techniques among live populations will be required. Up until recently, an assessment of stranded carcasses was the only basis used to determine the health of the dugong population. However, the findings of pathological studies involving harvested dugongs have rarely been documented due to the small sample size (*Owen, Gillespie & Wilkie, 2012*; *Woolford et al., 2015*). Due to the delays associated with the submission of postmortem reports, the collection of carcasses, and necropsy procedures, dugong health monitoring remains a significant challenge. Therefore, the study of the body and organ structure of the dugong has been significantly limited by the opportunistic sampling of stranded carcasses. Therefore, any studies conducted on the animal carcasses that have been recovered, would still be of value. To fully understand the postmortem outcomes, a better understanding of dugong physiology and illness response must be acquired. One of the primary obstacles to effective health screening has been the inadequate quantity of baseline clinical anatomy data regarding healthy dugongs (*Meager, 2016*). For these reasons, a knowledge of dugong anatomy and histology has become increasingly essential because its comprehension

**Table 1 The information of dugong carcass in the study.**

| Gender | Length (meters) | Weight (kilogram) | Condition |
|---|---|---|---|
| Female | 2.56 | 263.00 | Fresh carcass |
| Male | 1.20 | 34.50 | Fresh carcass |
| Male | 1.93 | 155.00 | Fresh carcass |
| Male | 1.32 | 57.00 | Fresh carcass |
| Female | 1.42 | 44.00 | Fresh carcass |
| Male | 2.70 | 313.50 | Fresh carcass |
| Male | 2.35 | 260.00 | Fresh carcass |
| Male | 1.5 | 27.00 | Fresh carcass |

would be necessary for gaining a fuller understanding of the medical and pathological processes.

Interestingly, the histological information obtained from various animals that are currently in danger of extinction, including elephants, white rhinoceros, leopard seals, and vaquita species, has been previously reported (*Gray, Canfield & Rogers, 2006*; *Plochocki et al., 2017*; *Rojas-Bracho et al., 2019*; *Woolford et al., 2015*). Unfortunately, histological information of normal tissue (non-pathological tissue) of the dugong is still lacking. This study has catalogued the histological characterizations for a stranded dugong carcass acquired from the Andaman Sea, Thailand. The results of this study can provide valuable insight into the limited available histological information of dugongs and will be beneficial for the monitoring of the health of living dugong populations.

# MATERIALS AND METHODS

## Sample collection

Eight dugong carcass (six males and two females), collected from the Andaman Sea of Thailand, were included in this study (Table 1). Necropsy results of some of the internal organs without pathological lesions and non-pathological tissue were included in this study. Four rotten carcasses were excluded from this study. Therefore, the remaining dugongs were deemed fit to serve as subjects of our microanatomy study. Tissue samples were collected from the carcass within 12 h after death and placed in 10% formalin. A total of 17 organ tissues were acquired from eight systems including:

1) Urinary system: kidney
2) Muscular system: striated skeletal muscle and smooth muscle
3) Cardiovascular system: cardiac muscle (ventricle), coronary artery, coronary vein
4) Respiratory system: Trachea (proximal part), lung (proximal, distal)
5) Gastrointestinal system: esophagus, stomach (fundus), small intestine (jejunum, ileum), liver, and pancreas
6) Reproductive system: testis

7) Lymphatic system: spleen and thymus

8) Endocrine system: pancreas

We conducted this study according to the guidelines of the Animals for Scientific Purposes Act, B.E. 2558 (2015). Some of our experiments were performed on the carcasses of dugong acquired from the Phuket Marine Biological Center, Phuket 83000, Thailand. Since no diagnostic procedure was employed to confirm the cause of death, no ethical approval was required for this study. This was further confirmed by the Animal Ethics Committee, Faculty of Veterinary Medicine, Chiang Mai University.

### Histological investigation

All non-pathological tissues received from the dugong carcass were fixed in 10% formalin for 24 h. The process employed for tissue preparation has been described in a previously published study (*Thitaram et al., 2018*). Individual non-pathological tissue samples were examined using a compound light microscope (Olympus BX53; Olympus, Tokyo, Japan) and program Slide Viewer 2.6. (3DHISTECH, Budapest, Hungary).

## RESULTS

### Urinary system

The kidney was divided into two parts: renal cortex and renal medulla (Figs. 1 and 2). Several glomeruli (condensed capillaries) were present in the renal cortex but not in the renal medulla. The renal capsule, a dense fibrous layer of collagen and elastic tissue, encapsulated the kidney (Fig. 1A). The glomerulus, proximal tubule, Henle's loop, distal tubule, and connecting tubule comprised the nephron, the smallest functional unit of the kidney. The arcuate vessels that were surrounded by connective tissue and smooth muscle tissue were identified at the cortico-medullary junction, or the area between the cortex and the medulla (Fig. 1A). The area of pars radiata was composed of straight tubules in the renal cortex, whereas the area of pars convolute was composed of convoluted tubules (Figs. 1A and 1B).

The proximal convoluted tubule had simple cuboidal epithelium with a brush border or microvilli, dark staining, and a round shaped nucleus, whereas the distal convoluted tubule did not have a brush border (Fig. 1D). The interlobular artery (Fig. 1C) is a branch of the renal arteries that supplies the renal parenchyma. It branches from the arcuate artery and supplies blood to each glomerulus *via* afferent arterioles. A high magnification cross section of the glomerulus complex revealed mesangial cells, podocytes (foot process cells), and glomerular capillaries (Figs. 1D–1F). The glomerulus has been divided into two parts: urinary (Fig. 1D) and visceral (Fig. 1E). The glomerulus was surrounded by the bowman's capsule, which was composed of simple squamous epithelium. Accordingly, ultrafiltrate fluid collects in the urinary space after being filtered through the glomerulus (Fig. 1F).

The renal medulla (Figs. 2A–2C) consisted primarily up of straight renal tubules that extend all the way through to the renal papilla. Straight tubules were divided into two types: thick and thin limbs. The ascending and descending limbs were thick with simple cuboidal epithelium. A segment of the Henle's loop with simple squamous epithelium

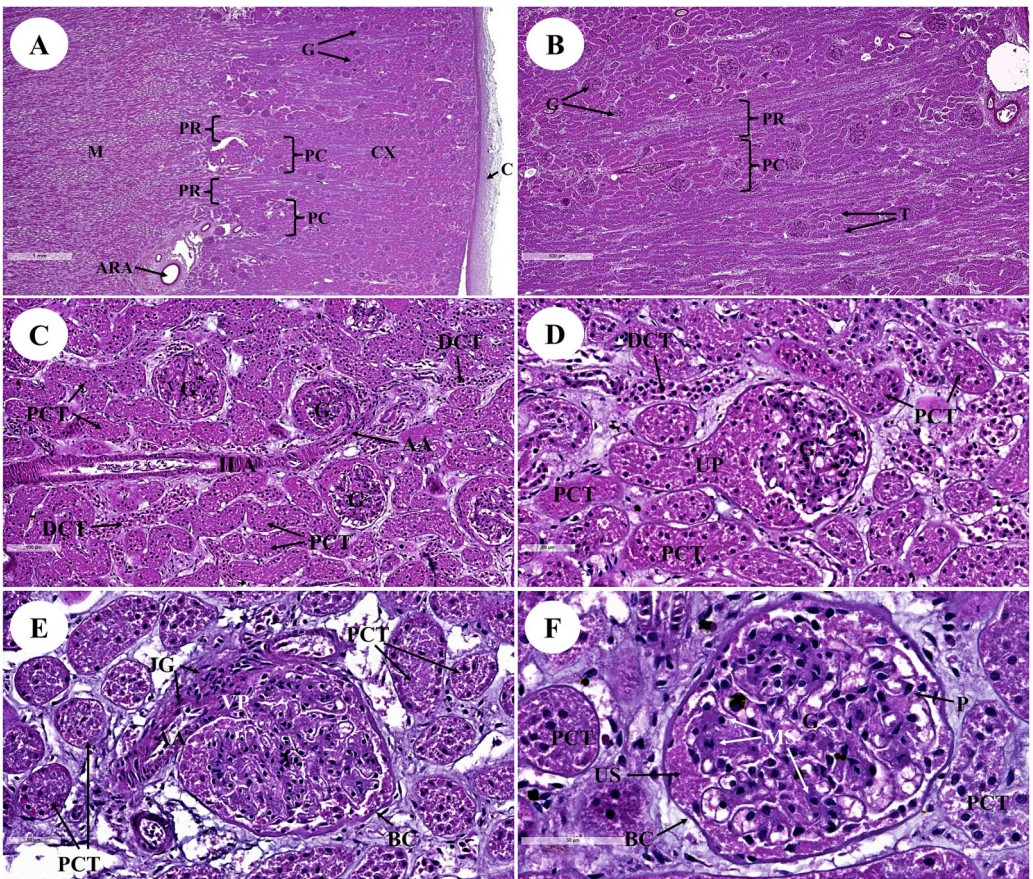

**Figure 1** Low and high magnification of histological sections of the renal cortex and medulla (A and B), renal tubules and renal vessel (C and D) and glomerulus complex (E and F). AA, afferent arteriole; ARA, arcuate artery; BC, bowman's capsule; C, renal capsule; CX, renal cortex; DCT, distal convoluted tubule; G, glomerulus; ILA, interlobular artery; JG, juxtaglomerular cell; M, renal medulla; MS, mesangial cell; PC, pars convolute; PCT, proximal convoluted tubule; PR, pars radiata; UP, urinary pole; US, urinary space; star shaped, podocyte cell. Hematoxylin and eosin staining. (A) 2×, (B) 5×, (C) 20×, (D and E) 40×, (F) 60×.

parallel to vasa recta or peritubular capillaries was identified. The collecting ducts provided simple columnar epithelium that may be seen clearly in the inner medulla (Fig. 2B).

The papilla duct or duct of Bellini (Fig. 2D) displayed high simple columnar epithelium, light staining, and a distinct boundary. This is where ultrafiltrate fluid gathered and was released into the renal pelvis *via* the cribriform area or the area of cribrosa.

## Gastrointestinal system

The gastrointestinal or digestive tract is comprised of a series of tubular organs including the esophagus, stomach, small intestine (consisting of the duodenum and duodenal diverticula, jejunum, and the ileum) and large intestine (consisting of the cecum, colon, and rectum). Additionally, accessory digestive organs are also present, namely the liver and the pancreas.

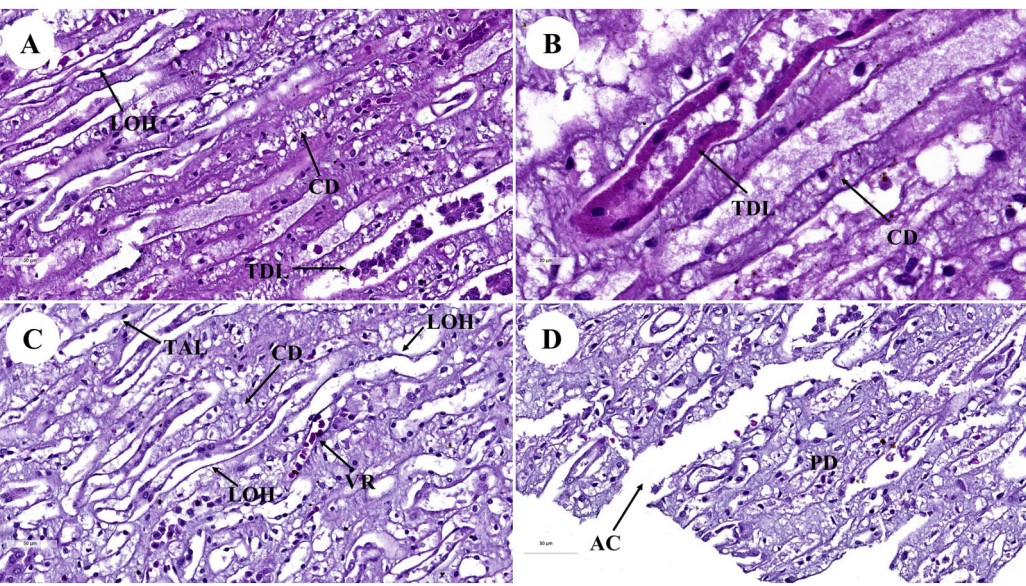

**Figure 2 Low and high magnification of histological sections of renal medulla; straight tubules (A and B), loop of Henle and collecting duct (C) and renal papilla (D).** AC, area of cribrosa; CD, collecting duct; LOH, thin segment Henle's loop; PD, papillary duct; TAL, thick ascending Henle's loop; TDL, thick descending; Henle's loop; VR, vasa recta. Hematoxylin and eosin staining. (A) 40×, (B) 100×, (C and D) 40×.

The four layers of the gastrointestinal tract extend from the luminal to the external surface including the tunica mucosa, tunica submucosa, tunica muscularis, and tunica serosa, respectively.

The esophagus is composed of four layers (Fig. 3A). The lumen of the esophagus is lined with a thick layer of non-keratinized stratified squamous epithelium (Fig. 3B). The lamina propria, which lies beneath the epithelium, is a dense irregular network of collagen fibers containing arterioles, venules, and capillaries. Compared to the tunica submucosa, the lamina propria has a denser connective tissue structure. The muscularis mucosae, or muscularis interna (Fig. 3C), consists of isolated bundles of smooth muscle tissue located between lamina propria and tunica submucosa. The loose connective tissue of the tunica submucosa contains numerous blood vessels and small parasympathetic ganglia that are scattered throughout, forming the submucosal (Meissner) plexus (Fig. 3D). Within the tunica muscularis (or muscularis externa), there are clearly visible inner circular and outer longitudinal layers of smooth muscle tissue, separated by connective tissue and parasympathetic ganglion cells of the myenteric (Auerbach) (Fig. 3E). The outermost layer is the tunica adventitia, which is comprised of loose connective tissue containing blood vessels, lymphatic vessels, and nerves (Fig. 3F).

In the stomach, the fundic gland region of the stomach exhibited typical features found in monogastric animals (Fig. 4A). The surface epithelium was lined with simple columnar epithelium, and the lamina propria contained straight branched tubular gastric glands (Fig. 4B). Parietal cells with the eosinophilic cytoplasm and chief cells with basophilic cytoplasm are the predominant cell types in this gland region (Fig. 4C). The muscularis

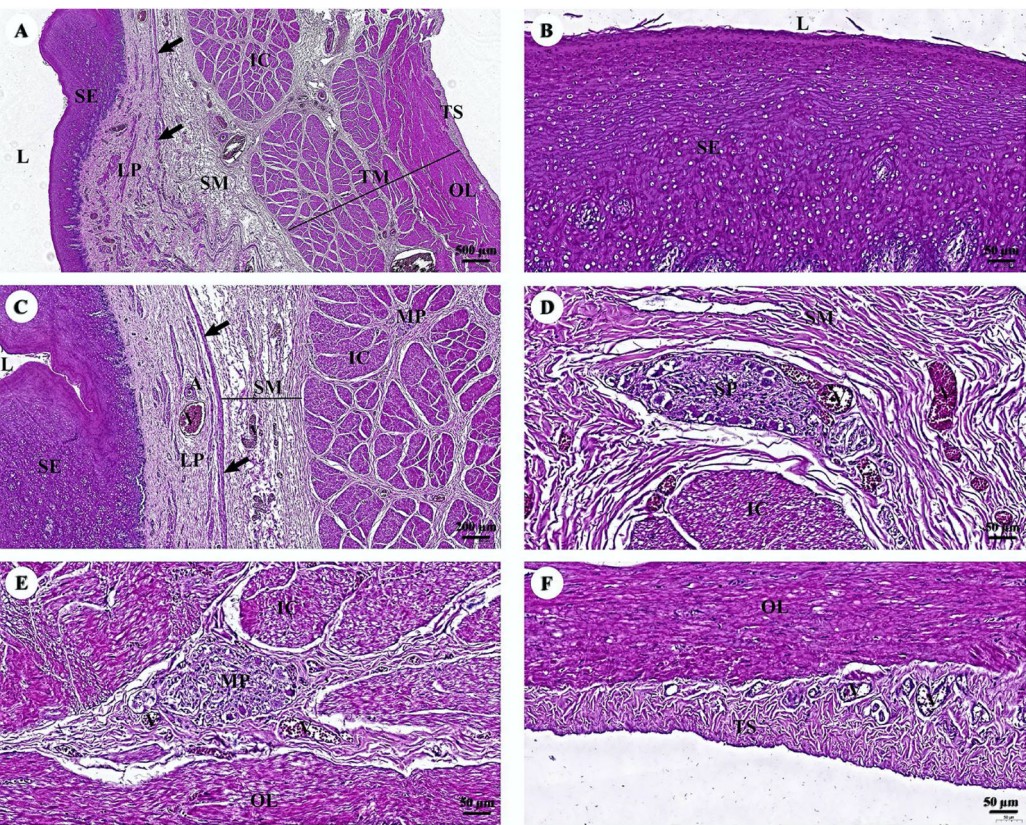

**Figure 3 (A–F) Light microscopy micrographs at different magnifications of the esophagus.** Study sites: arrow, muscularis mucosae; A, arteriole; IC, inner circular smooth muscle; L, lumen; LP, lamina propria; MP, myenteric nerve plexus; OL, outer longitudinal smooth muscle; SE, surface epithelium; SG, submucosal mixed gland; SP, submucosal nerve plexus; SM, tunica submucosa; TM, tunica muscularis; TS, tunica serosa; V, venule. Hematoxylin and eosin staining.

mucosae, a continuous layer of smooth muscle tissue, separates lamina propria and tunica submucosa. The tunica submucosa of the stomach is relatively dense and distensible, consisting of collagen fibers, larger blood vessels, lymphatic vessels, submucosal nerve plexuses, and other free cells. The tunica muscularis is broad and comprised of three layers of smooth muscles: inner oblique, middle circular, and outer longitudinal, which are arranged differently and separated by thin layers of connective tissue. The myenteric nerve plexus is located between the middle and outer smooth muscle layers (Fig. 4D). The tunica serosa consists of loose connective tissue, adipose tissue, arterioles, and venules. At this magnification, the serosal lining, which covers the peritoneal surface, is thin and barely visible.

Two parts of the small intestine, the jejunum (Fig. 5) and ileum (Fig. 6), were examined. The tunica mucosa of the jejunum is lined by a typical simple columnar intestinal epithelium with goblet cells. The villi are short and irregular in outline. The jejunum is characterized by large circumferentially oriented submucosal folds known as plicae circulares, which are distinct and numerous (Figs. 5A and 5D), although the mucosal surface displayed autolysis. The muscularis mucosae varies from a thin layer to more

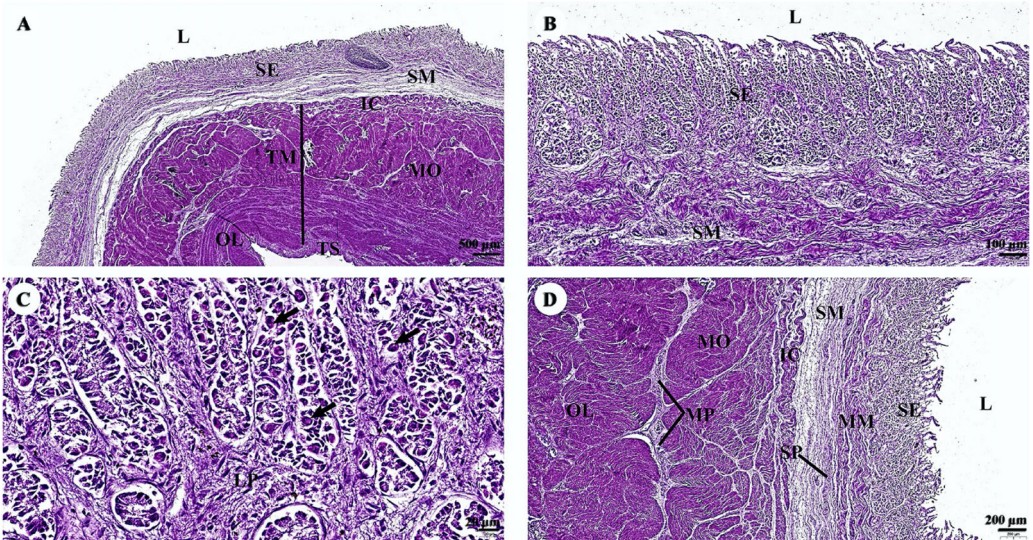

**Figure 4** (A–D) Light microscopy micrographs at different magnifications of the stomach (Fundus). Study sites: arrow, parietal cell; IC, inner circular smooth muscle; L, lumen; LP, lamina propria; MM, muscularis mucosae; MO, middle oblique smooth muscle; MP, myenteric nerve plexus; OL, outer longitudinal smooth muscle; SE, surface epithelium; SM, tunica submucosa; SP, submucosal nerve plexus; TM, tunica muscularis; TS, tunica serosa; hematoxylin and eosin staining.

extensive bundles of smooth muscle tissue, clearly delineating the delicate lamina propria from the underlying submucosa. Generally, the submucosa is thick and contains extensive amounts of collagen, a vascular network, nerves, and a remarkable submucosal nerve plexus (Fig. 5B). The myenteric nerve plexus is prominent and situated between the inner circular and outer longitudinal layer of smooth muscle tissue in the tunica muscularis (Fig. 5C). The tunica serosa is comprised of a slim layer of loose connective tissue covered by mesothelium. Interestingly, numerous smooth muscle bundles were found within the submucosa (Fig. 5D, arrows).

The surface epithelium of the ileum consists of a simple straight tubular gland lined by simple columnar epithelium with intraepithelial goblet cells (Fig. 6A). The submucosa is separated from the lamina propria by the muscularis mucosae, which lies immediately beneath the mucosal crypts. The intestinal crypts of the ileum were deeper when compared with those of the jejunum (Fig. 6B). Prominent clusters of subepithelial non-encapsulated lymphoid tissue, known as Peyer's patches, were observed in the submucosa (Fig. 6C). The tunica muscularis and the tunica serosa resemble those of the jejunum (Fig. 6D).

The outer surface of the liver lobe was covered by the liver capsule, which was composed of dense regular collagenous connective tissue and covered by a layer of mesothelial cells emanating from the visceral peritoneum (Fig. 7A). Liver parenchyma exhibited a lobular architecture that contained portal tracts and central or terminal hepatic venules. Portal triads, or portal areas, consisted of one or more branches of the hepatic artery, portal vein, and bile ductule, and were surrounded by a network of connective tissue (Fig. 7B). The hepatocytes were arranged in rows or hepatic plates, separated by sinusoids lined by

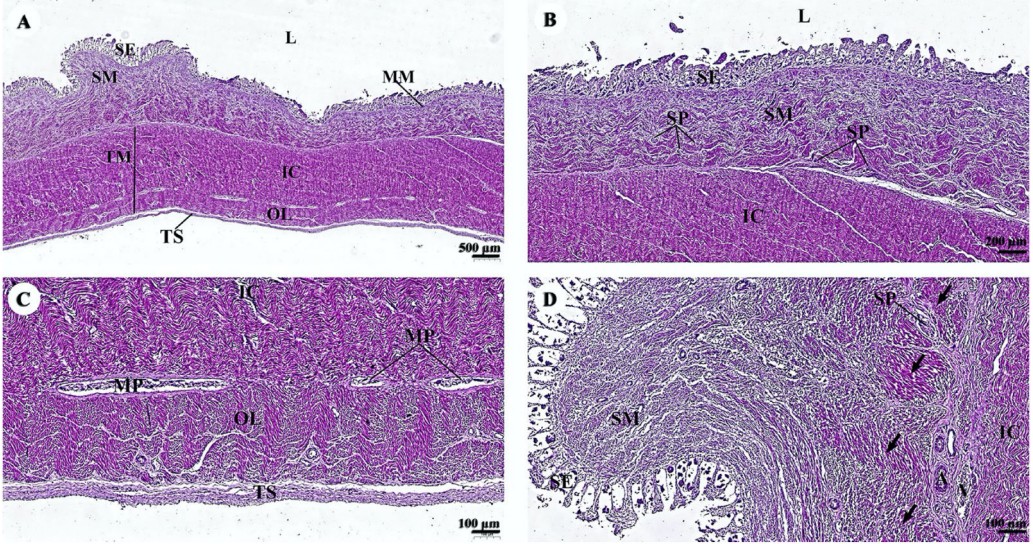

**Figure 5 (A–D) Light microscopy micrographs at different magnifications of the small intestine (jejunum).** Study sites: arrows, smooth muscle; A, arteriole; IC, inner circular smooth muscle; L, lumen; MM, muscularis mucosae; MO, middle oblique smooth muscle; MP, myenteric nerve plexus; OL, outer longitudinal smooth muscle; SE, surface epithelium; SM, tunica submucosa; SP, submucosal nerve plexus; TM, tunica muscularis; TS, tunica serosa; V, venule. Hematoxylin and eosin staining.

endothelial cells. Sinusoidal blood typically drains into central veins (Fig. 7C). The bile ductule within the portal areas were lined by a simple cuboidal epithelium (Fig. 7D).

The pancreas was covered by collagenous connective tissue, which extended into the parenchyma as septa. These septa divide parenchyma into lobules. Blood vessels, nerve fibers, and ganglia run within the interlobular connective tissue septa (Fig. 8A). Assorted sizes of islets of Langerhans, which are endocrine tissue of the pancreas, were scattered throughout the exocrine tissue (Fig. 8B). The pancreatic acinus was composed of pyramid-shaped secretory epithelial cells. Spindle-shaped centroacinar cells were located at the central lumen (Fig. 8C). These cells were extensions of the intercalated duct into each pancreatic acinus. Intercalated ducts, which are the smallest ducts in the pancreatic ductal system, drained into small intralobular ducts lined by simple cuboidal epithelium (Fig. 8D).

## Cardiovascular system

The cardiovascular system consists of the heart, arteries, veins, and capillaries. Under low magnification (1×), the cross-sectional area of the heart revealed three layers: the epicardium, the outermost layer of the heart's wall; the myocardium, the middle layer; and the endocardium, the innermost layer, which covers all of the internal surfaces of the heart chambers (Fig. 9A). The mesothelium, a single layer of flattened epithelial cells covering the epicardium, was observed at a low magnification (10×). Additionally, this layer was connected to the myocardium by layers of connective tissue, while a thick layer of adipose tissue was implanted with coronary arteries, coronary veins, and autonomic nerve fibers (Fig. 9B). Due to the complex spiral arrangement in which the cardiac muscle fibers were

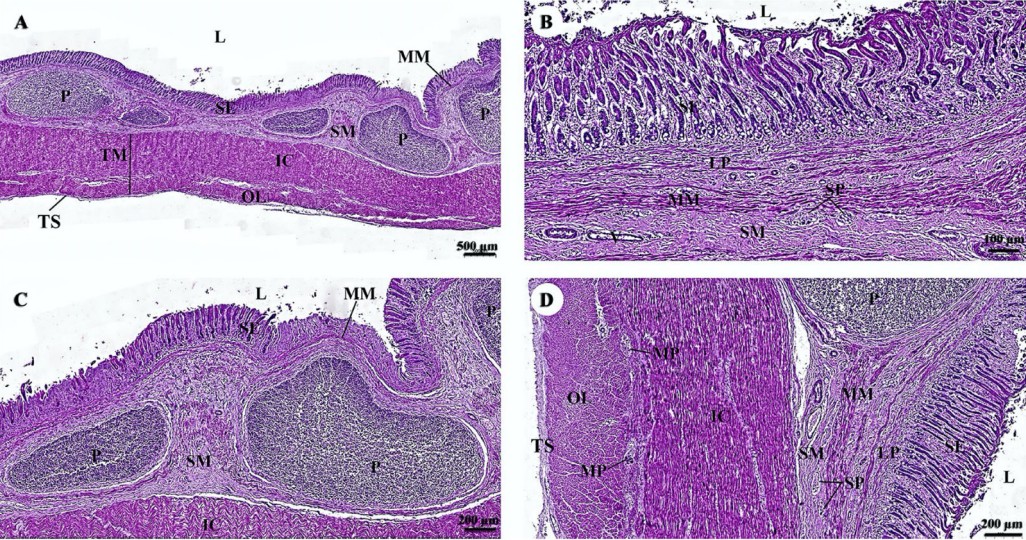

**Figure 6 (A–D) Light microscopy micrographs at different magnifications of the small intestine (ileum).** Study sites: IC, inner circular smooth muscle; L, lumen; MM, muscularis mucosae; MO, middle oblique smooth muscle; MP, myenteric nerve plexus; OL, outer longitudinal smooth muscle; P, Peyer's patch; SE, surface epithelium; SM, tunica submucosa; SP, submucosal nerve plexus; TM, tunica muscularis; TS, tunica serosa; V, venule. Hematoxylin and eosin staining.

arranged in the myocardium, cross-, oblique-, and longitudinal sections of cardiac muscle bundles were observed at a low magnification (10×) (Fig. 9C). At a low magnification (20×) of the cross-sectional area of the myocardium layer of the ventricle, bundles of cardiac muscle fibers were bound together with perimysium, which is thin connective tissue made primarily of reticular fibers and frequently includes blood vessels (arteries, veins), lymphatic vessels, and nerves (Fig. 9D). At a high magnification (40×) of the cross-sectional area, an oval-shaped nucleus was observed in the center of the cardiac muscle fibers (cardiomyocytes), while fibroblasts were observed in the lateral border of the cell (Fig. 9E).

As was observed in a longitudinal section of the cardiac muscle fiber cells, they displayed extensively branched architecture with an ovoid nucleus in the middle of the cell. Moreover, their ends were connected to one another by intercalated discs (Fig. 9F). In addition, an intercellular collagen network and numerous capillary networks surrounding the cardiac muscle fibers were observed (Figs. 9E and 9F). At a low magnification (20×), the endocardium revealed a single layer of flattened epithelium (endothelium) covering the inner surface of the cardiac chamber. The layer beneath the endocardium that bound the endocardium to the myocardium was identified as the subendocardial layer, which contained loose connective tissue, blood vessels, nerves, and Purkinje fibers (Purkinje cardiomyocytes). As has been observed in cardiac muscle fibers, Purkinje fibers contain cross striations, while intercalated discs were observed at the junctions of the two fibers. However, they differed from typical heart muscle fibers in size and thickness (larger than cardiac muscle fibers by 2–4 times). Moreover, one or more large nuclei centrally located within the fibers were observed. Compared to normal cardiac

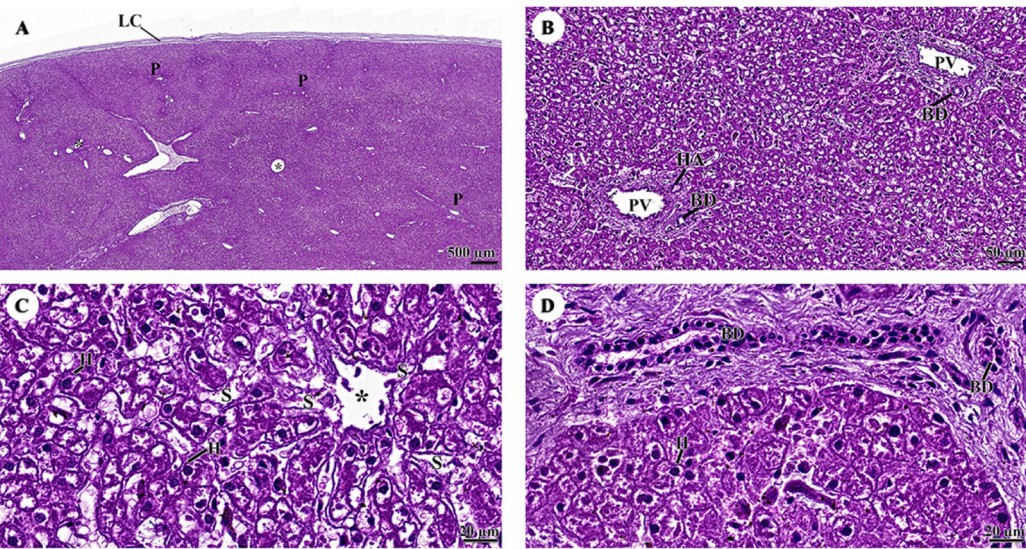

**Figure 7 (A–D) Light microscopy micrographs at different magnifications of the liver.** Study sites: asterisk, central vein; BD, bile ductule; H, hepatocyte; HA, hepatic artery; LC, liver capsule; P, portal triads; PV, portal vein; S, sinusoid; Hematoxylin and eosin staining.

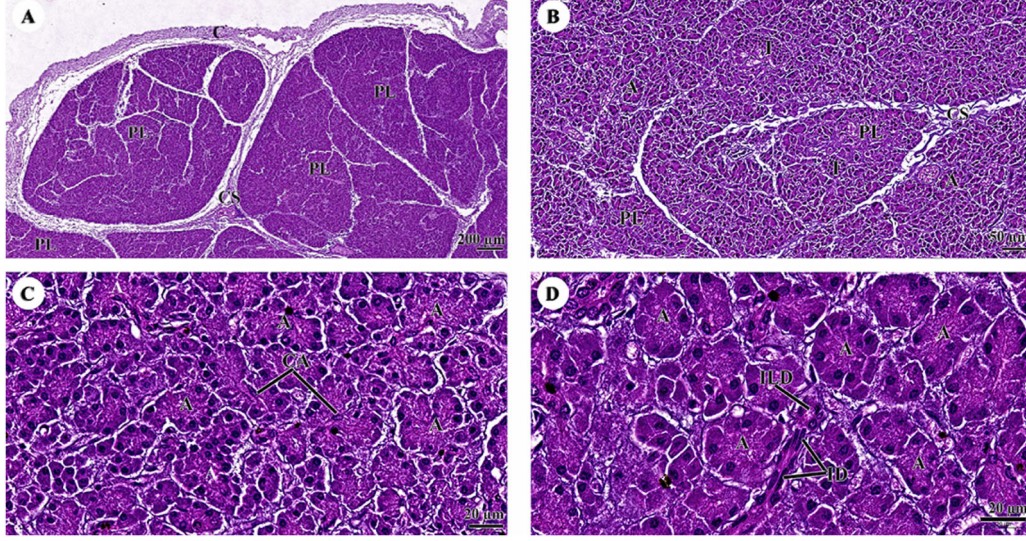

**Figure 8 (A–D) Light microscopy micrographs at different magnifications of the pancreas.** Study sites: A, acinus; C, capsule; CA, centroacinar cell; CS, connective tissue septum; I, islet of Langerhan; ID, intercalated duct; ILD, intralobular duct; PL, pancreatic lobule; Hematoxylin and eosin staining.

muscle cells, the cytoplasm of the Purkinje fibers was more lightly stained (light pink) than the cytoplasm of normal cardiac muscle cells (pink or red). Furthermore, Purkinje fibers had a lower density of capillary networks surrounding their fibers than the cardiac muscle fibers (Fig. 9G).

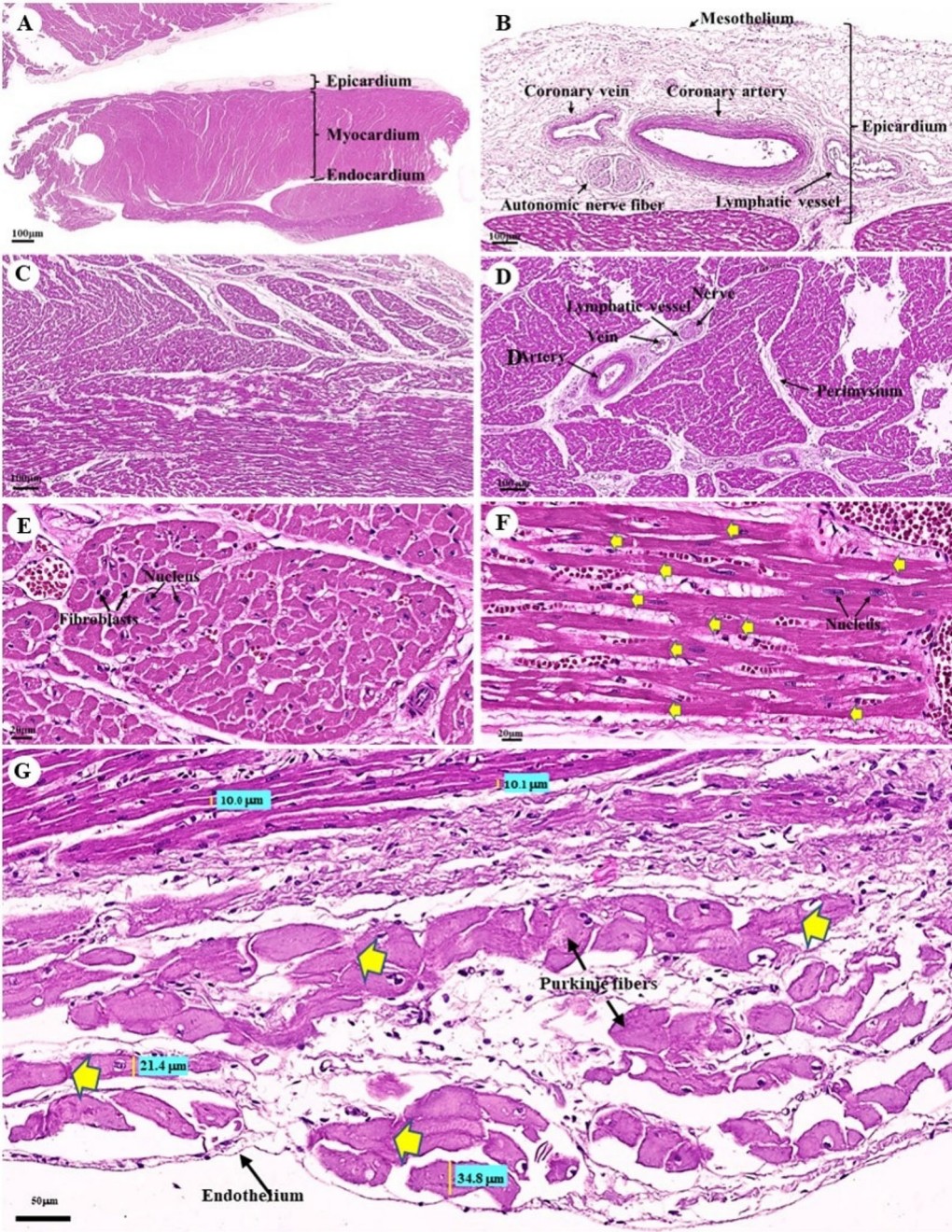

**Figure 9 Low and high magnification (1× to 40×) of histological sections of cardiac muscle from ventricle (A), epicardium of heart (B), myocardium layer of heart (C), cardiac muscle bundle in cross section (D and E), cardiac muscle fibers in longitudinal section (F) and endocardium of heart (G).** Yellow arrow represented intercalated discs. Hematoxylin and eosin staining.

The tunica intima, tunica media, and tunica adventitia were found in the layers of all blood vessels, except for the capillaries. Arteries are characterized as round and smaller in diameter than veins because of their elasticity (Figs. 10A–10E). Under a low magnification (15×), at the epicardial layer of the heart, the cross-sectional area of the coronary vessels,

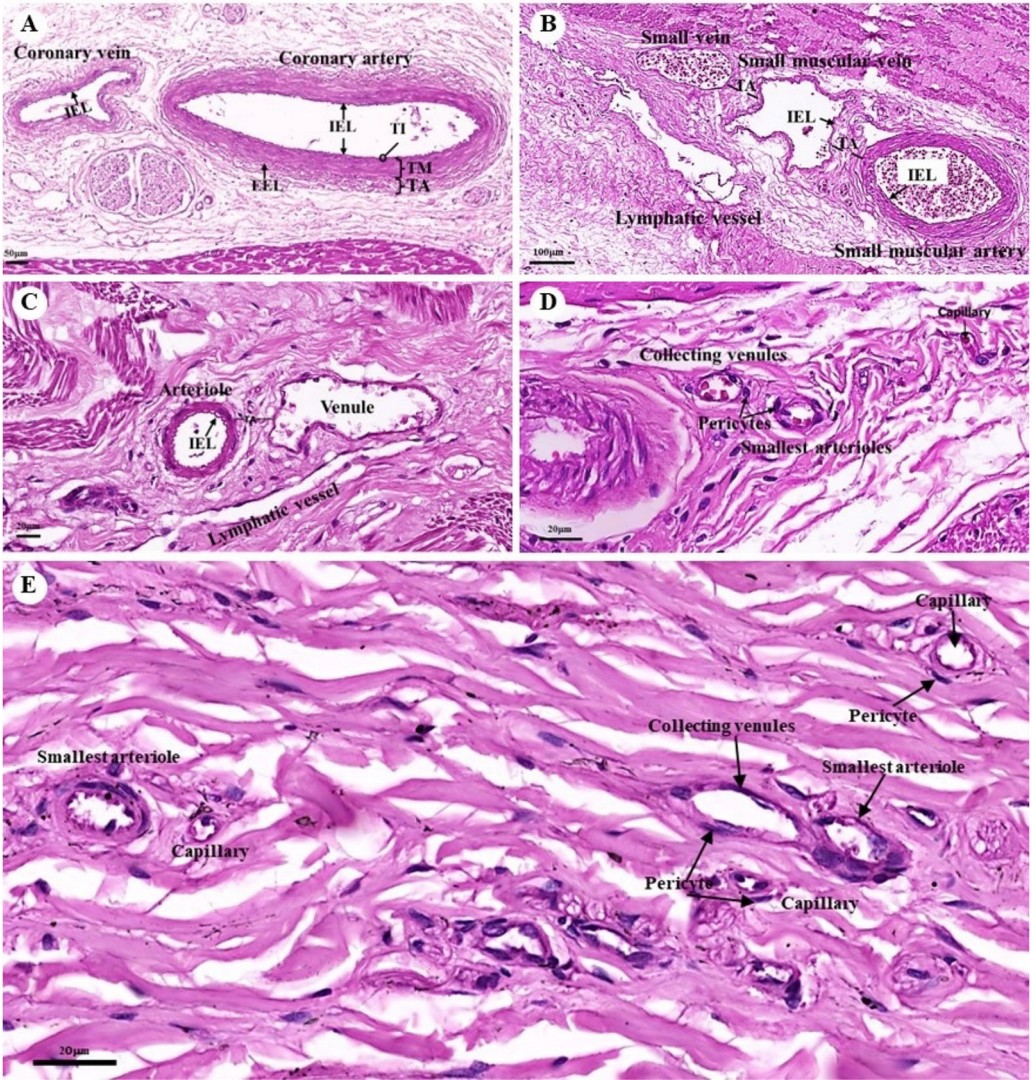

**Figure 10 Low and high magnification (15× to 40×) of histological sections of the coronary vessels (coronary artery and coronary vein) (A), small muscle artery and small muscle vein (B), arterioles and venules (C), the various diameters of arterioles, venules, and capillaries of the general tissue organ (D) and he smallest arterioles, collecting venules and capillaries (E).** Study sites: IEL, internal elastic lamina; EEL, external elastic lamina; TA, tunica adventitia; TI, tunica intima; TM, tunica media. Hematoxylin and eosin staining.

the coronary artery (epicardial artery), and the coronary vein (epicardial vein), were demonstrated. The coronary artery, also known as a muscular artery, is the thickest layer of the tunica media (the middle layer). It is primarily composed of smooth muscle cells and elastin, while internal and external elastic laminae (IEL and EEL) were stained black and thus easily visible. The coronary veins have been shown in the same three layers as the coronary arteries, but boundaries were indistinct and external elastic laminae were not observed. Because of the thinner tunica media that appeared with less smooth muscle tissue and elastic fibers, the coronary veins appeared to have collapsed (Fig. 10A). At a low magnification (15×), small muscle arteries and small muscle veins (Fig. 10B) were seen to

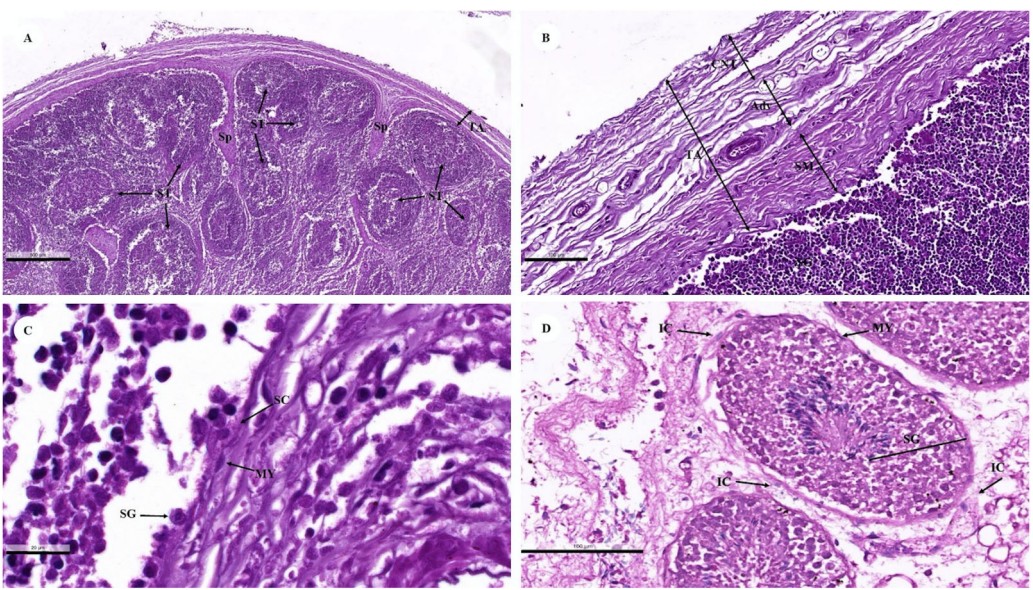

**Figure 11 Low (A and B) and high (C and D) magnification of histological sections of a juvenile dugong testis.** TA, tunica albuginea; Sp, septa; ST, seminiferous tubules; CNT, connective tissue; Adv, adipose tissue and blood vessel; SM, smooth muscle; SG, spermatogenic cell; SC, Sertoli cell; MY, myoid cell; IC, interstitial cell.

travel in pairs and share tunic adventitia with one another. At a high magnification (40×), arterioles and venules (Fig. 10C) were observed to do the same. In addition, the internal elastic lamina could be seen in the walls of the small muscular arteries, small muscular veins, and arteriole. But external elastic laminae were not observed. Additionally, the walls of the small muscular arteries, small muscular veins, and arteriole could all be shown to have internal elastic lamina. However, no external elastic laminae were observed.

At a high magnification (40×) of the general tissue organ, various diameters of the arterioles, venules, and capillaries could be observed (Figs. 10D and 10E). The smallest arterioles lacked an internal elastic lamina, with the endothelium resting directly on the basement membrane. In addition, their tunica media contained only a single layer of smooth muscle fibers, and the tunica adventitia was not observed. The wall of the collecting venules was composed of a complete layer of pericytes serving as the capillaries and a thin, continuous endothelium. However, their luminal diameter was 3–5 times greater than that of the capillaries.

## Reproductive system
### Testis

A histological examination of the testes revealed that the dugongs were juvenile and mature males. Each juvenile testis was surrounded by a tunica albuginea (Fig. 11A) characterized by distinctive layers: inner smooth muscle, middle adipose, and vascular, and outer dense irregular connective tissue layers (Fig. 11B). The tunica albuginea extended into the testis *via* connective tissue septa (Fig. 11A) that divided the lobules containing the seminiferous tubules (Fig. 11A). The seminiferous tubules were outlined by a thick

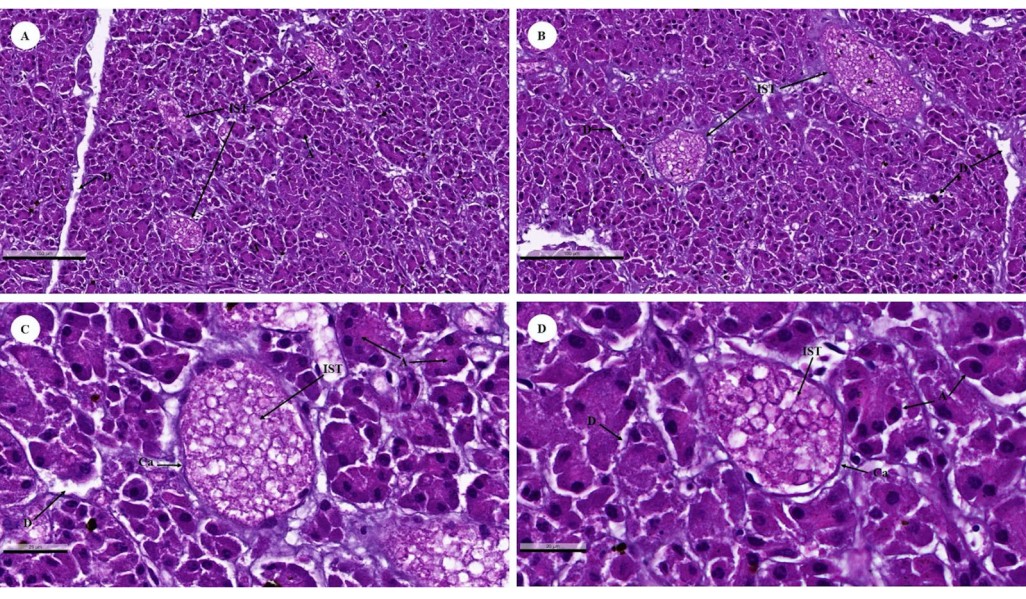

**Figure 12 Low (A and B) and high (C and D) magnification of histological sections of a juvenile dugong pancreas.** IST, islet of Langerhans; A, acinar cell; D, duct; Ca, capsule.

basement membrane (smooth muscle) with myoid cells (Fig. 11C). A stratified layer of spermatogenic and Sertoli cells lay within the tubules (Figs. 11C and 11D). Sertoli cells were visible in the basolateral aspect of the seminiferous tubules as large irregularly shaped cells with a pale nucleus, with primary spermatogonia that have darker, rounder nuclei. Primitive spermatids were found over the apical end of the Sertoli cell, which then underwent spermiogenesis and budded off into the lumen of the seminiferous tubules (Figs. 11C and 11D). The only spermatogenic cells in the juvenile dugong calf testes were spermatogonium, while neither spermatocytes nor spermatids were observed (Fig. 11A). In the adult male dugongs, spermatogenic cells appeared in various stages, including spermatogonium, spermatocytes, and spermatids (Fig. 11D). Between the tubules, small clusters of interstitial tissue containing Leydig cells were observed (Fig. 11D). These were polygonal cells with a large round nucleus and prominent nucleolus, vacuolized eosinophilic cytoplasm, blood vessels, and high lipid content (Fig. 11D).

## Endocrine system
### *Endocrine pancreas*
The islets of Langerhans were composed of groups of secretory cells supported by a fine collagenous network containing numerous fenestrated capillaries (Figs. 12A and 12B). Each islet was surrounded by a delicate capsule (Figs. 12C and 12D). The endocrine cells were small with a pale-stained granular cytoplasm. By contrast, large cells of the surrounding exocrine pancreatic acini were stained strongly (Figs. 12A–12D). The endocrine islets of Langerhans were round or ovoid in shape and of varying size (Figs. 12A–12D).

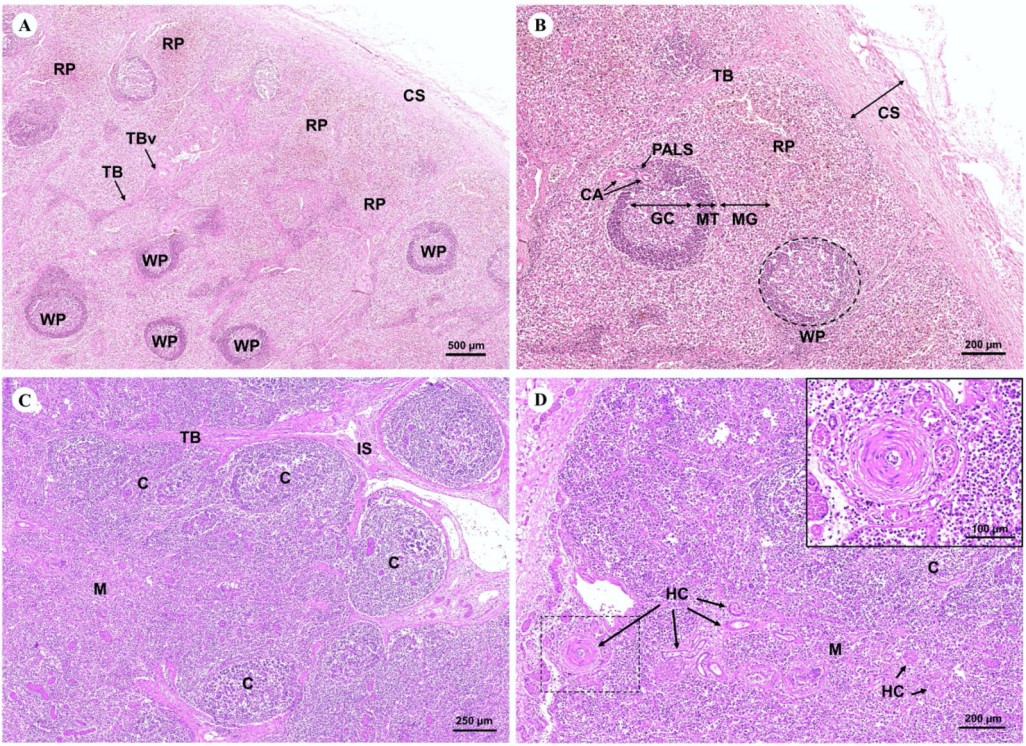

**Figure 13 Low and high magnification of microanatomy of the spleen (A and B) and the thymus (C and D).** Study sites: CS, capsule; RP, red pulp; WP, white pulse; GC, germinal center; MT, mantle zone; MG, marginal zone; TB, trabecula; TBv, trabecular vessel; CA, central arteries; C, cortex; M, medulla; IS, interlobular septum; SL, splenic lobules; HC, Hassall's corpuscles.

## Lymphatic system

The spleen, a large lymphoid organ without a cortex/medulla structure, was surrounded by a dense connective tissue capsule that extended inward as splenic trabeculae through to the trabecular vessel. The splenic parenchyma was divisible into the red and white pulp (Fig. 13A). The red pulp consisted of a erythrocytes-filled reticular meshwork and sinusoids and splenic cords. The white pulp consisted of lymphoid nodules and periarteriolar lymphoid sheaths (PALS) surrounding central arteries. The lymphoid nodule was subdivided into a germinal center, mantle zone, and marginal zone (Fig. 13B).

The thymus, the primary lymphoid organ, was divided into incompletely separate lobules. Each lobule was made up of a darkly peripheral cortex and a lightly inner medulla (Fig. 13C). In the thymic medulla, epithelial cells congregated into Hassall's corpuscles, which were exclusively found in the thymus gland (Fig. 13D).

## Muscular tissue

Muscular tissue has been found in the muscularis externa layer of the dugong's internal organs (Figs. 14A and 14B). Two types of muscular tissues were observed including the visceral striated muscle in the esophagus (Figs. 15A, 15C, and 15E) and the smooth muscle in the stomach (Figs. 15B, 15D, and 15F). For the visceral striated muscle, the fascicle was

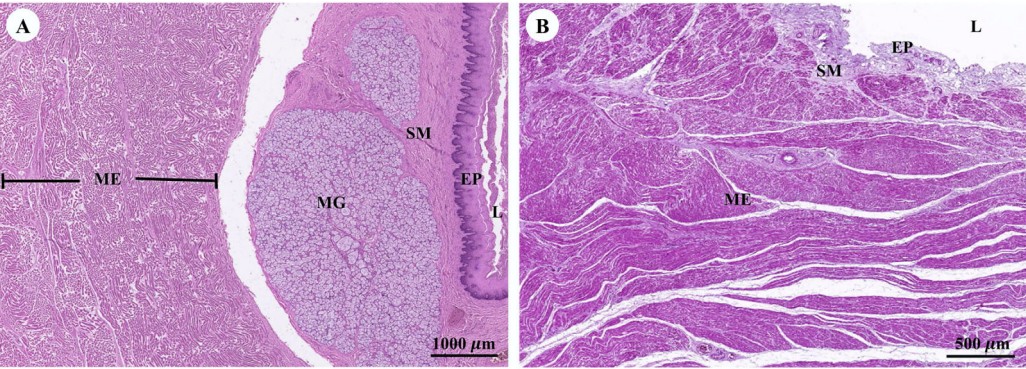

**Figure 14 Low magnification of histological section of muscular tissue found in muscularis externa layer of dugong's esophagus (A) and stomach (B).** Abbreviations: EP, epithelium; L, lumen; ME, muscularis externa; MG, mucosal gland; SM, submucosa. Hematoxylin and eosin staining.

formed from the many muscle fibers surrounding the perimysium (Fig. 15A). While each muscle fiber was covered by endomysium, blood vessels could also be found between these covered structures (Fig. 15C). At higher magnifications, a cross striation was observed on the muscle fiber (Fig. 15E), while the nucleus was located at the peripheral (Figs. 15C and 15E). In contrast, the structure of the smooth muscle was different from the visceral striated muscle. A group of smooth muscle cells were separated from each other by connective tissue (Fig. 15B). A high magnification also revealed the smooth shape of the muscle cells in the cross section, which could be seen as round cells with a central nucleus (Fig. 15D), while fusiform-shaped muscle cells were observed in the longitudinal section without cross striations (Fig. 15F).

## Respiratory system

The trachea (Fig. 16A) represented an incomplete hyalin cartilage structure with the perichondrium, while chondrocytes, a longitudinal muscle (the trachealis muscle) was found around the trachea in the lacunae. In the respiratory epithelium, a pseudostratified ciliated epithelium was observed with goblet cells. In the submucosal layer, tracheal mixed glands were observed. In the bronchus (Fig. 16B), gobleted and ciliated cells were found around the bronchial respiratory epithelium. Bronchial hyalin cartilages were presented in many pieces of cartilage plates. The bronchiole (Figs. 16C and 16D) is a respiratory airway without a cartilage structure. It can be divided into two parts, the terminal bronchiole and the respiratory bronchiole. The terminal bronchiole respiratory epithelium had two cell types, namely goblet cells and non-ciliated cuboidal Clara cells. The function of the goblet cells is to produce mucous secretions, while Clara cells produced a pulmonary surfactant, reserved stem cells, and contained detoxification enzymes to protect the lower respiratory tract when exposed to oxidative substances. The respiratory bronchiole is characterized by a terminal branch of alveolar ducts, wherein each respiratory bronchiole divides further into several alveolar ducts that have numerous alveoli opening along their lengths.

An intra-lung structure (Fig. 17A) was found to be comprised of three types of airway structures composed of a secondary bronchus that was present in the large lumen and

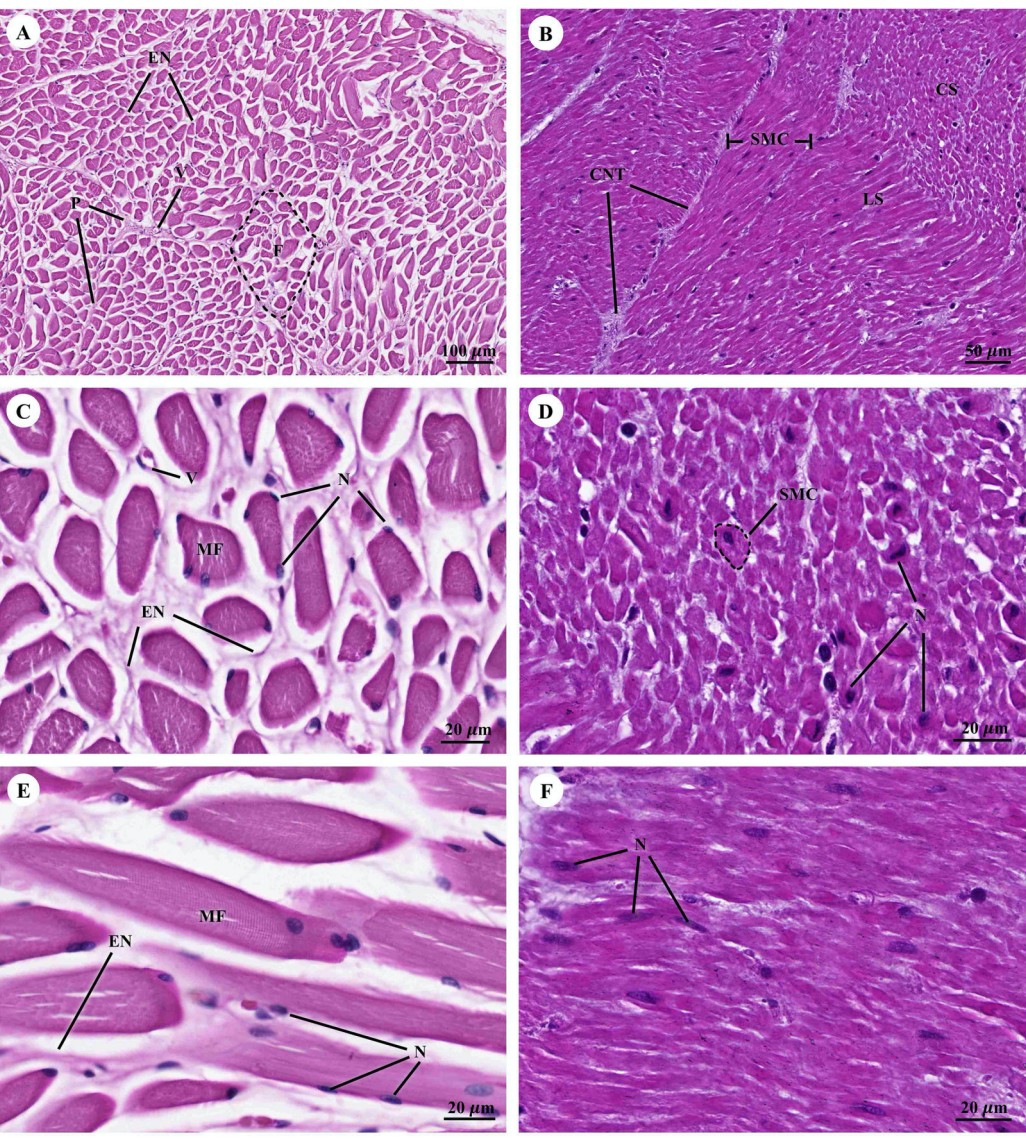

**Figure 15 Low and high magnification of histological section of muscular tissue from visceral striated muscle of esophagus (A, C and E) and smooth muscle of stomach (B, D and F).** Abbreviations: V, blood vessel; CNT, Connective tissue; CS, cross section; EN, endomysium; F, fascicle; LS, longitudinal section; MF, muscle fiber; N, nucleus; P, perimysium; SMC, smooth muscle cell. Hematoxylin and eosin staining.

numerous cartilage plates. Tertiary bronchus was present but smaller than the secondary bronchus, which had a small cartilage plate. A terminal bronchiole was present in the smallest lumen without being surrounded by cartilage. The visceral pleura (Fig. 17B) was found to be composed of a thick outer layer of lung structures present in dense fibrous connective tissue with a few blood vessels. The alveolar lining (Fig. 17C) was found to contain numerous terminal bronchioles and a few cartilage plates of bronchus, namely the terminal bronchiole, respiratory bronchiole, alveolar duct, and alveolar sac, respectively. At a higher magnification, the alveolar lining (Fig. 17D) was represented by type I

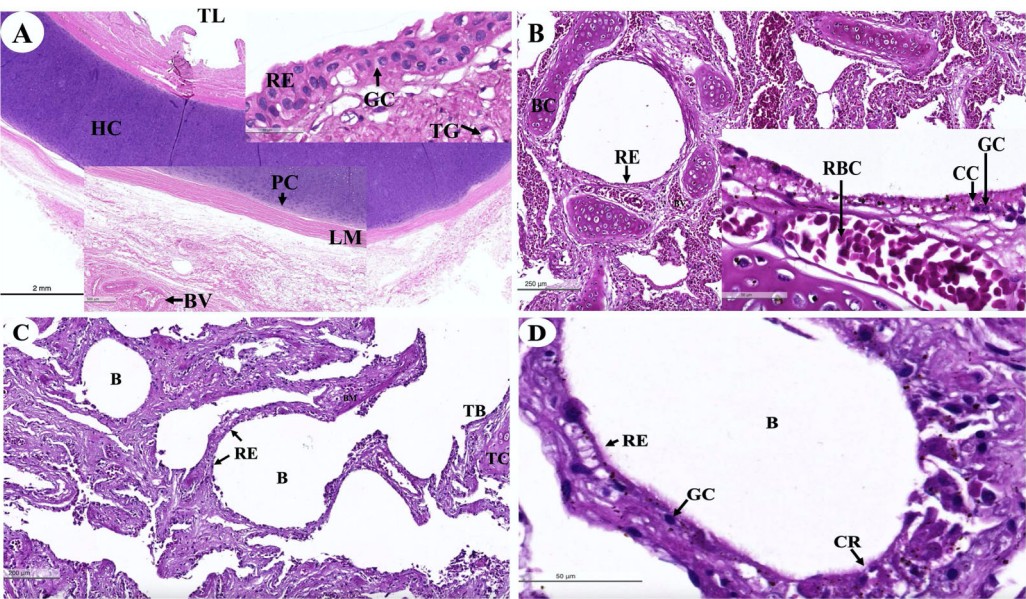

**Figure 16 Low and high magnification of histological sections of the trachea (A), bronchus (B), bronchiole (C and D).** Study sites: B, bronchiole; BC, bronchial cartilage; BV, blood vessel; CC, ciliated cell; CG, goblet cell; CR, Clara cell; HC, hyalin cartilage; LM, longitudinal muscle; PC, perichondrium; RB, respiratory bronchiole; RBC, red blood cells; RE, respiratory epithelium; TB, terminal bronchiole; TC, terminal bronchiole cartilage; TG, tracheal gland; TL, tracheal lumen. Hematoxylin and eosin staining.                                   

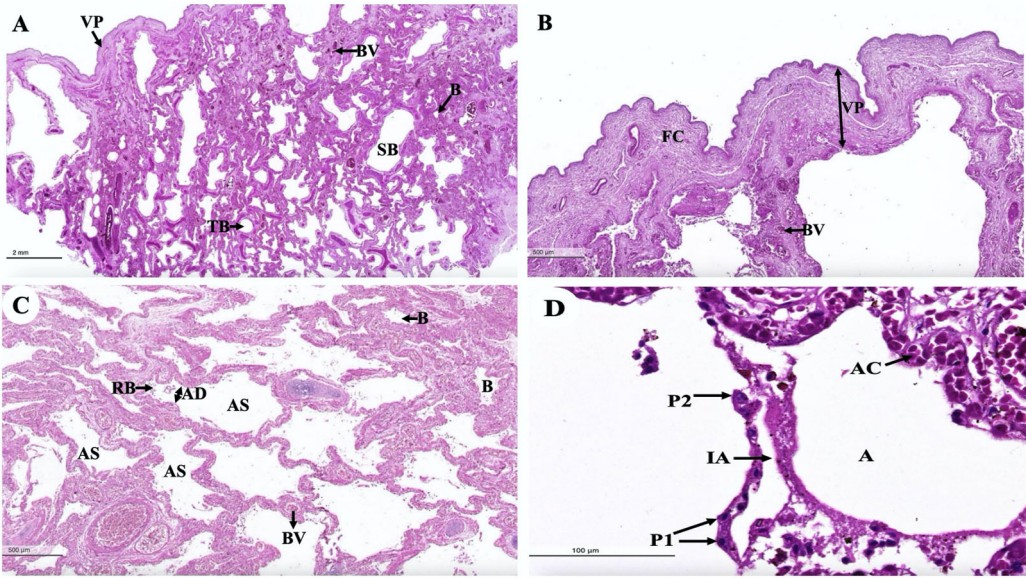

**Figure 17 Low and high magnification of histological sections of the intra-lung structure (A), visceral pleura (B), alveolar lining (C and D).** Study sites: A, alveoli; AC, alveolar capillary; AD, alveolar duct; AS, alveolar sac; B, bronchiole; BV, blood vessel; FC, fibrous connective tissue; IA, interalveolar septum; P1, pneumocyte type I; P2, pneumocyte type II; RB, respiratory bronchiole; SB, secondary bronchus; TB, tertiary bronchus; VP, visceral pleura. Hematoxylin and eosin staining.                                   

pneumocytes (squamous cells) and type II pneumocytes (round cells). The intra-alveolar septum contained dense connective tissue and an alveolar capillaries network.

## DISCUSSION

With regard to the urinary system, the kidneys of many marine animals are specialized in reniculate form (multi-lobed), with each lobe (renula) having all the components of a complete metanephric kidney (*Vardy & Bryden, 1981*). Although it is unknown why marine animals have reniculate kidneys, the fact that some large terrestrial mammals also have reniculate kidneys has led to speculation that they represent an adaptation that is associated with their enormous body size (*Williams, 2006*). The kidneys of marine animals are frequently larger than those of terrestrial mammals of comparable body mass (*Beuchat, 1996*). Both the dugong and the manatee are marine mammals (*Berta, Sumich & Kovacs, 2015*). Furthermore, unlike manatees, dugongs appear to be physiologically independent of fresh water (*Berta, Sumich & Kovacs, 2015*). That could be why dugong kidneys differ from manatee kidneys (*Berta, Sumich & Kovacs, 2015*).

Furthermore, the kidney of the dugong differs from all other marine mammals in that its kidneys were non-lobular and smoothly elongated on the exterior, resulting in a long medullary crest and implying some type of transverse medullary lobulated kidney (*Ca, 2002*). This kidney structure is similar to that of camels (*Abdalla & Abdalla, 1979*), horses (*Bolat et al., 2013*), swine (*Sampaio, Pereira-Sampaio & Favorito, 1998*), and humans (*Wallace, 1998*). Dugong kidneys, on the other hand, were both grossly and microscopically different from elephant kidneys (*Maluf, 1995*) and all other marine mammals, including manatees (*Maluf, 1989*), leopard seals (*Gray, Canfield & Rogers, 2006*), pinnipeds (seals and sea lions) (*Stewardson et al., 1999*; *Vardy & Bryden, 1981*), and cetaceans (whales and dolphins) (*Maluf & Gassman, 1998*; *Pfeiffer, 1997*), all of which have externally lobulated or multi-reniculated kidneys. The remarkable differences in renal anatomy and histology observed across sirenians imply that these differences in kidney appearance may be attributed to differences in the habitats of each species (*Ortiz, 2001*).

The surface was covered with a thick and strong fibrous renal capsule consisting of collagen and elastic fibers that are thicker than they are in canines (*Bulger, Cronin & Dobyan, 1979*), Indian buffaloes (*Ommer & Mariappa, 1970*), and elephants (*Maluf, 1995*; *Thitaram et al., 2018*). The arcuate arteries were found at the cortico-medullary junction and included connective tissue and smooth muscle. Notably, this feature was similar to what has been reported in a previous study on the leopard seal (*Cave & Aumonier, 1964*). Furthermore, the cribriform area was flat or concave near the end of the renal papilla, which is an enlarged opening of papillary ducts to the renal pelvis. Accordingly, this feature is similar to that of both the pygmy hippopotamus (*Maluf, 1994*) and the elephant (*Maluf, 1995*; *Thitaram et al., 2018*).

With regard to the gastrointestinal system, the morphology and histomorphology of the dugong gastrointestinal tracts (stomach, duodenum, and duodenal diverticula) had previously been described by *Kenchington (1972)* and *Murray et al. (1977)*. Their findings indicate that the dugong is a nonruminant herbivore. Microscopically, the structure of the gastrointestinal tracts of the dugong and the manatee (*Reynolds & Krause, 1982*; *Reynolds*

& *Rommel, 1996*) exhibits significant similarities. The esophageal mucosa of the dugong is characterized by a non-keratinized stratified squamous epithelium, resembling that of the elephant (*Van Aswegen et al., 1994*). In contrast, hind gut fermenters, such as horses, rabbits, and guinea pigs, typically possess a keratinized stratified squamous type of esophageal mucosa, while the degree of keratinization is known to be dependent upon the type of food ingested (*Samuelson, 2007*). Unlike the elephant esophagus, which contains numerous submucosal mixed glands secreting mucus to lubricate and protect the luminal surface (*Thitaram et al., 2018*), the dugong esophagus lacks submucosal glands.

The stomach of the dugong is a simple mammalian stomach lined with glandular mucosa. Notably, the pyloric epithelium of the dugong lacks stratified squamous cells, which differentiates it from the Florida manatee. In terms of innervation and vascular supply, myenteric nerves and blood vessels are abundant in the muscularis externa of the dugong, albeit to a lesser extent than in the manatee (*Reynolds & Rommel, 1996*). Among the sections of the small intestine, the duodenum of the dugong closely resembles that of the manatee, another member of the order Sirenia (*Reynolds & Krause, 1982*). Furthermore, the ileum of the dugong exhibits numerous Peyer's patches, a feature shared with other Sirenians and terrestrial mammals. The cytoarchitecture of the liver and pancreas in the dugong demonstrates a normal organization when compared with what has been observed in other terrestrial or marine mammals.

With regard to the cardiovascular system, a histological investigation of the cardiovascular system in the organs of dugongs was conducted to compare our outcomes with the findings of studies involving humans, other domestic mammals, and birds. As far as we know, no previous research has been conducted on the histological structure of the dugongs' cardiovascular system. Previous studies have focused on the anatomical structure of the heart of the dugongs (*Rowlatt & Marsh, 1985*). In a histological study of the organs of dugongs, many organs were closely examined, but the heart, adrenal, thyroid, and pituitary glands were not examined (*Woolford et al., 2015*). In this study, the heart of the dugong could be divided into three layers consisting of the epicardium, myocardium, and endocardium, which is similar to the hearts of humans, other domestic mammals, and birds (*Bacha & Bacha, 2012*; *Liebich, 2020*). Interestingly, we found that the epicardial layer of the dugong contained a thick layer of epicardial fat, or epicardial adipose tissue, that was composed of adipocytes, nerve tissue, and coronary vessels (Fig. 1B). Moreover, the layer may also include immune cells (*Iacobellis & Bianco, 2011*). There are noticeable species-specific differences in epicardial adipose tissue; in healthy humans, it can cover up to 80% of the surface of the heart (*Iacobellis et al., 2005*) and account for 20% of the total ventricular weight, in contrast to rodents, where epicardial adipose tissue is almost absent (*Corradi et al., 2004*). The thickening of the epicardial adipose tissue might be related to many roles in cardiac physiology, such as the storage of free fatty acids (FFAs) as a source of energy for the heart, protection of the myocardium from high fatty acid levels and associated lipotoxicity, thermoregulation in heat production and protection of the heart against cold, mechanical protection of the heart and coronary arteries against the torsion of the arterial pulse wave and cardiac contraction, and immunological support by their immune cells that help to protect the heart against pathogens and inflammatory activators

(*Antonopoulos & Antoniades, 2017*). Moreover, increased epicardial fat thickness has been found to be related to obesity, body mass index, the metabolic syndrome, and heart disease, which are characterized by inflammation, hypertension, and disturbances in insulin sensitivity (*Wu et al., 2017*).

The cardiac muscle microstructure of dugongs was similar to that of humans, other domestic mammals, and Asian elephants (*Bacha & Bacha, 2012*; *Liebich, 2020*; *Thitaram et al., 2018*; *Zhang, 1999*), but was mostly similar to the horse and Asian elephants in the way that their ventricles contained numerous groups of muscle fascicles (Fig. 1D) (*Bacha & Bacha, 2012*; *Liebich, 2020*; *Thitaram et al., 2018*). Hematoxylin and eosin staining demonstrated the space surrounding the cardiac muscle fibers in a cross sectional view (Fig. 1E) and a longitudinal sectional view (Fig. 1F). Importantly, there are a number of points of interest: (1) they could be fixation artifacts, cell shrinkage gaps left behind when the tissue was fixed in formalin or another fixative, or they could be areas where the tissue was somewhat pulled apart during sectioning; (2) at the time of tissue collection (*Kachaeva & Shenkman, 2012*); (3) this space consisted of an excessive accumulation of intercellular collagen networks and numerous capillary networks that could have been present due to the aging associated with myocardial collagen accumulation or fibrotic remodeling (*Horn & Trafford, 2016*); or (4) these could be the result of muscle atrophy (*Aughey & Frye, 2001*).

The Purkinje fibers of the dugong were distributed in the subendocardial connective tissue in the ventricle, similar to those of humans and other domestic mammals (*Bacha & Bacha, 2012*; *Liebich, 2020*; *Zhang, 1999*). Hematoxylin and eosin staining indicated that the microstructure of the dugongs' Purkinje fibers were similar to that of humans and other domestic mammals (*Bacha & Bacha, 2012*; *Liebich, 2020*; *Zhang, 1999*). Intercalated discs at the junction of two fibers also resembled those of human and monkey Purkinje fibers (*Zhang, 1999*; *Zhang et al., 1996*). The histological findings stained with hematoxylin and eosin demonstrated that the microstructure of the dugongs' blood vessels, including the coronary arteries and veins, small muscle arteries and veins, arterioles and venules, and continuous capillaries, were similar to those of humans and other domestic mammals (*Bacha & Bacha, 2012*; *Liebich, 2020*; *Zhang, 1999*).

The limitations of the histological studies of the cardiovascular system naturally include the lack of some blood vessels, including the aorta, vena cava, pulmonary artery, and pulmonary vein; while the elastic artery, large vein, medium artery, and medium vein were not examined, respectively. The results of the current studies indicate that almost all of the cardiovascular organs of dugongs in this study had histological findings that were similar to those of other mammal species and birds, with the exception of the dugongs' epicardial layer, which thickened more than that of other mammalian species and birds. These characteristics may have been linked since they are both marine mammals that live in cold water, and if their habitat is cold, it is crucial for thermoregulation—the process of producing heat and shielding the heart from the cold. Their neutrinos and metabolism may also play a role in this process.

Regarding the reproductive system, aspects of the male dugong reproductive system have been described; however, a complete description of the microanatomy was not possible because the males in this study were juveniles and mature males. This study was

conducted to investigate the reproductive system of male dugongs, but only testis organs were collected. Overall, the microanatomy revealed similarities to that of other species, such as the Florida manatee (*Trichechus manatus latirostris*) (*Hernandez et al., 1995*; *Perez, 2015*), African elephant (*Loxodonta africana*) (*Short, Mann & Hay, 1967*), and common dolphin (*Delphinus delphis*) (*Murphy, Collet & Rogan, 2005*). The tunica albuginea of most domestic mammals is composed of dense irregular connective tissue, with the exception of the stallion, boar, and ram, which are composed primarily of smooth muscle tissue (*Bacha & Bacha, 2000*). Dugong testes contained an inner layer of smooth muscle in the tunica albuginea, whereas this layer was located in the outer portion in the stallion (*Bacha & Bacha, 2000*). Thus, the structure of the dugong tunica albuginea more closely resembled that of domestic livestock species. Adult manatees have a thicker tunica layer than juveniles and calves (*Perez, 2015*), but because we only had juvenile testes with a complete tunica albuginea layer, but no mature males with a tunica albuginea layer, an evaluation of how the tunica albuginea may change with age was not possible. Thus, additional samples across age groups are needed to more fully understand testicular changes in dugongs. Peritubular myoid cells, which are the primary cellular components of the wall of seminiferous tubules and exhibit smooth muscle-like characteristics, have been identified in all mammalian species, although organization within the peritubular interstitial tissue varies among species (*Maekawa, Kamimura & Nagano, 1996*; *Zhou et al., 2019*). Similar to dugongs, laboratory rodents, including rats, hamsters, and mice, have only one layer of myoid cells in the testis (*Preen, 1995*). It is believed that myoid cells play an important role in the intratesticular transport of immobile sperm and so are integral to spermatogenesis (*Zhou et al., 2019*).

Interstitial cells, or Leydig cells, are the primary source of androgens and play a crucial role in numerous vital physiological processes in males, such as sperm production, sexual development, and the maintenance of secondary sexual characteristics and behaviors (*Zhou et al., 2019*). Previous studies have suggested that the gonadal activity of mature male dugongs is variable, so not all males in a given population produce spermatozoa continuously (*Kwan, 2002*; *Marsh, 1995*; *Marsh, Heinsohn & Marsh, 1984*). Male dugongs in the tropics were found to have spermatogenic testes mainly in the latter half of the year (austral winter to spring), which is consistent with a diffused breeding season (*Kwan, 2002*; *Marsh, 1995*; *Marsh, Heinsohn & Glover, 1984*). However, the timing of male reproductive cycles and the extent of inactivity have not been exhaustively studied, as only carcass analyses have provided information on individual males at the time of death.
The spermatogenic cycle of dugongs is similar to that of Florida manatees, elephants, and hyraxes (*Perez, 2015*). While testicular function in both dugongs and manatees is influenced by season, these species are comparable in terms of the body length at which sexual maturity occurs (*Perez, 2015*). Any disparity could be due to species-specific differences in life history and genetics or could be linked to differences in resource availability and the stability of their native habitats.

The exact age and length of each dugong in this study was unknown, as only tissue samples were available, but it was presumed there were juvenile and mature male testis. This type of information provides important life history data that could have an impact on

dugong captive management. Additional tissue analysis and other reproductive system organs are required, especially to better define sexual maturity age and how spermatogenic activity changes over time and with age. Future studies should also include testing hormone concentrations in the blood, feces, or urine in addition to analyzing functionality of the seminiferous tubules and associated cells. Controlling the spermatogenic cycle is largely dependent upon FSH and testosterone (*Sharpe, Maddocks & Kerr, 1990*), although stress can also have direct and indirect effects on testosterone production. Thus, it may be useful to evaluate correlations between testicular stage, testosterone, and glucocorticoid concentrations to better understand how to manage dugong reproduction. Finally, it is important to examine serial sections of testes, preferably from breeding-season carcasses, to determine if spermatogenic activity is uniform throughout the gonads (*Hernandez et al., 1995*).

With regard to the endocrine system (pancreas), the exocrine component of the pancreas was composed of numerous tubuloacinar secretory units. The purpose of this research study was to understand more about the male dugong's endocrine system. Unfortunately, in this study, we could only use the pancreas organs obtained from these carcasses to identify this system. The endocrine islets of Langerhans (pancreatic islets) are clusters of epithelial cells that are dispersed throughout the secretory units (*Bacha & Bacha, 2012*; *Eurell & Frappier, 2006*). In addition to being organized into islets, the endocrine cells are scattered individually and in small clusters among the exocrine acini (*Fowler & Mikota, 2006*). The pancreas of the dugong is composed of exocrine and endocrine tissue and enclosed by a thin connective tissue capsule that is rich in adipose tissue, blood vessels, and fine collagen fibers. This feature is similar to that of other mammalian species such as the leopard seal, horse, cattle, donkey, and sheep (*Gray, Canfield & Rogers, 2006*; *Hafez, Zaghloul & Caceci, 2015*; *Mahesh et al., 2017*), as well as the gazelle (*Hamza, 2018*). The histological structure of the endocrine islets of the Langerhans in the dugong was similar to that of other mammals in that they are spherical or ellipsoid in shape and of varying overall size, with irregular shapes reflecting the pressure of an adjacent structure, often a duct, or were limited by a tissue plane (*Gray, Canfield & Rogers, 2006*; *Hafez, Zaghloul & Caceci, 2015*; *Longnecker, 2021*; *Mahesh et al., 2017*).

The pancreas regulates blood sugar concentrations, while isolated groups of pale-staining islet cells are dispersed throughout the secretory units. Histochemistry or transmission electron microscopy was used to distinguish four distinct cell types (*Eurell & Frappier, 2006*). The majority of cells were A, alpha cells that secrete glucagon (a polypeptide hormone secreted in response to hypoglycemia or growth hormone stimulation); or B, beta cells that secrete insulin into the bloodstream in response to a rise in concentration of blood glucose or amino acids. Rare D, or delta, cells that release somatostatin, while F cells secrete pancreatic polypeptide. These cells belong to the group responsible for amine precursor uptake and decarboxylation (APUD) (*Aughey & Frye, 2001*). Importantly, gap junctions join the islet cells to facilitate intercellular communication (*Eurell & Frappier, 2006*).

Comparable to other animal species, the endocrine pancreas of the dugong contained numerous types of secretory cells; however, in the H & E stained preparations, the cell

types were indistinguishable, necessitating the use of special staining techniques. For example, autolytic and inadequately preserved pancreatic tissues prevented identification of endocrine components in Asian elephants (*Thitaram et al., 2018*). Dolphin islets are primarily formed of beta-cell cords that are dispersed throughout the islet. Few islets predominantly had beta-cells in their center cores (*Colegrove & Venn-Watson, 2015*). In comparison, a typical rat pancreas shows normal Langerhans islets with pale spherical and ovoid-beta cells in the center (*Saad et al., 2015*). As a result, future research must use specialized techniques such as immunohistochemistry to identify cell types in the pancreas of dugongs, while other organs in the endocrine system, including the thyroid gland and adrenal gland, should be collected to better understand endocrinology in dugongs.

With regard to the lymphatic system, the histology of the dugong spleen displayed no differences from the spleen of other mammals. The dense connective tissue capsule and trabeculae consisted of smooth muscle tissue, as well as elastic and collagen fibers. The splenic parenchyma was unable to be divided into a cortex/medulla structure like other lymphoid organs. However, it had two functionally different regions: red pulp and white pulp or the splenic follicle. Interestingly, the follicles of the dugongs in this study were much larger than the follicles of the Australian dugong, but a prominent germinal center or hyperplastic follicle were not observed (*Woolford et al., 2015*). The white pulp and red pulp were found all over the spleen including in the branch of the splenic microcirculation; trabecular vein and artery, and central arteries (*Mescher, 2018*).

The histological findings of the thymus taken from a dugong in this study was similar to the manatee (*Goldbach, 2010*) and other mammals (*Cave & Aumonier, 1967*; *Igbokwe & Ezenwaka, 2017*). The structural component of the thymus consisted of a connective tissue capsule, as well as a cortex, medulla, and Hassall's corpuscles. The mammalian thymus is gradually replaced by connective and adipose tissue after age-related thymic involution (*Igbokwe & Ezenwaka, 2017*). However, a few fat invasions in the thymic parenchyma of the dugong were observed, which was in agreement with a study conducted on a dugong from Bahrein (*Cave & Aumonier, 1967*), suggesting a unique pattern of thymic involution in the dugong. In summary, the histology of the spleen in dugongs revealed no significant differences when compared with other mammalian spleens. Nevertheless, further studies are required to understand the functional significance of the unique thymic histology observed in dugongs.

With regarding to the muscular system, the overall histological structure of the muscular tissue of the dugong was similar to that of other marine mammals including cetaceans (*Sierra et al., 2015*; *Suárez-Santana et al., 2020*) and other closely related species, namely the manatee (*Grossman, Hamilton & De Wit, 2014*; *Reynolds & Rommel, 1996*), and the Asian elephant (*Thitaram et al., 2018*). In this study, the diameter of the visceral striated muscle found in the esophagus of the dugong was at the same size of the striated muscle in the body of the vocal cord of the Florida manatee by cross section in the histological study (*Grossman, Hamilton & De Wit, 2014*). In the previous study, the visceral striated muscle had also been found in the esophagus of the Asian elephant (*Thitaram et al., 2018*); however, minor differences were found when compared with the
muscular tissue of the dugong in this study. Lesser values in the density and diameter of muscle fibers were observed in the Asian elephant's esophagus, which was also the case for an elephant clave (*Thitaram et al., 2018*). This might be because of differences in the span of age, just as an increase in muscle fiber size, could commonly occur in older animals (*Miljkovic et al., 2015*).

Although muscular tissue had been observed in many organs of the dugongs in this study, including the intestinal tract, kidney, heart and trachea, knowledge pertaining to the muscular tissue of the skeleton muscle that attaches to the tendon and bone from the limbs and the body core has still been limited in both morphological and histological studies, suggesting that this organ should be included in the further studies.

With regard to the respiratory system, the *Dugong dugong* and *Dugong manatee* are two species in the family of Sirenia. This study was conducted to investigate the respiratory structure of the *Dugong dugong* in comparison to other marine species, marine mammals, and terrestrial mammals by histological evaluation. We hypothesize that the respiratory adaptations for diving in dugongs are similar to those of other marine mammals (*Pabst, Rommel & McLellan, 1999*). It was found to be similar to the tracheal structure of certain marine mammals, such as dolphins, with their continuous cartilage rings and absence of trachealis muscle (*Bagnoli et al., 2011*). The trachea of dugongs was found to possess an incomplete cartilage ring and present a trachealis muscle that was similar to that of terrestrial mammals. The tracheal respiratory epithelium was pseudostratified ciliated columnar epithelium with submucosal goblet cells. The bronchus structure had multi-plated hyaline cartilages that were similar to other terrestrial mammals. Surprisingly, the terminal bronchioles of the dugong appear to have a small amount of cartilage around the terminal bronchiole lumen. In a previous study of Sirenian, it was found that manatees are also reinforced with cartilage (*Pabst, Rommel & McLellan, 1999*). The terminal bronchiole cartilage structure indicated an increase in general resistance and stiffness in comparison with terrestrial mammals (*Bagnoli et al., 2011*). The respiratory epithelium of the bronchioles was composed of pseudostratified ciliated columnar epithelium with goblet cells. In the respiratory bronchioles and alveolar ducts, there was a thick wall that was supported by smooth muscle tissue surrounded by collagen and elastin fibers. The visceral pleura represents thick fibrous connective tissue with a few networks of blood vessels.

The respiratory physiology of dugongs has indicated that inspiration time was close to animals with paired nostrils and forced expiration, as is commonly present in animals that need to replenish oxygen storage. The dugong has 2–3 min breath intervals, but can extend as long as 24 min in individuals during resting time (*Reynolds, 1981*). Hypoxemia and hypercapnia conditions have been reported in relation to the respiratory drive of Amazonian manatees. It was found that carbon dioxide, rather than oxygen, was concluded to be the primary controller of ventilation and drive activity.

In summary, the different anatomy, histology, and stiffer mechanism properties of the dugong respiratory tract might be considered an adaptation of reinforcement and resistance to diving under a high-pressure environment. In comparisons with certain terrestrial mammals, such as cats, dogs, or pigs, this could be explained by respiratory

airway structure differences and performance limits displayed by terrestrial *vs* marine mammals during breath-holding diving.

## CONCLUSIONS

In this study, we have described the normal histological outcomes for different tissues of the dugong. Almost all structures were similar to those of other reported mammal species, whereas some tissues differed from other mammalian species and manatees. Therefore, histological information obtained from various organs in this study could provide an essential database for future microanatomical studies and play an essential role in diagnosing sick dugongs or those associated with an unknown cause of death. However, more samples from both sexes and various ages would enhance the important microanatomical information for this endangered species.

## ACKNOWLEDGEMENTS

The authors thank the Phuket Marine Biological Center, Phuket 83000, Thailand for providing us with necessary samples.

### Funding

This work was supported by Chiang Mai University through the Research Administration Office, which provided a budget to Research Center for Veterinary Biosciences and Veterinary Public Health, Chiang Mai University, Chiang Mai 50200, Thailand.
The funders had no role in study design, data collection and analysis, decision to publish, or preparation of the manuscript.

### Grant Disclosures

The following grant information was disclosed by the authors:
Chiang Mai University through the Research Administration Office.
Research Center for Veterinary Biosciences and Veterinary Public Health.
Chiang Mai University, Chiang Mai, Thailand: 50200.

### Competing Interests

Korakot Nganvongpanit is an Editor for PeerJ (Veterinary Medicine). The other authors authors declare that they have no competing interests. The authors alone are responsible for the content and writing of the article.

### Author Contributions

- Patcharaporn Kaewmong performed the experiments, prepared figures and/or tables, authored or reviewed drafts of the article, and approved the final draft.
- Pathompong Jongjit performed the experiments, prepared figures and/or tables, authored or reviewed drafts of the article, and approved the final draft.
- Araya Boonkasemsanti performed the experiments, prepared figures and/or tables, authored or reviewed drafts of the article, and approved the final draft.

- Kongkiat Kittiwattanawong performed the experiments, prepared figures and/or tables, authored or reviewed drafts of the article, and approved the final draft.
- Piyamat Kongtueng performed the experiments, prepared figures and/or tables, authored or reviewed drafts of the article, and approved the final draft.
- Pitchaya Matchimakul analyzed the data, prepared figures and/or tables, authored or reviewed drafts of the article, and approved the final draft.
- Wasan Tangphokhanon analyzed the data, prepared figures and/or tables, authored or reviewed drafts of the article, and approved the final draft.
- Prapawadee Pirintr analyzed the data, prepared figures and/or tables, authored or reviewed drafts of the article, and approved the final draft.
- Jaruwan Khonmee analyzed the data, prepared figures and/or tables, authored or reviewed drafts of the article, and approved the final draft.
- Songphon Buddhasiri analyzed the data, prepared figures and/or tables, authored or reviewed drafts of the article, and approved the final draft.
- Promporn Piboon analyzed the data, prepared figures and/or tables, authored or reviewed drafts of the article, and approved the final draft.
- Sonthaya Umsumarng analyzed the data, prepared figures and/or tables, authored or reviewed drafts of the article, and approved the final draft.
- Raktham Mektrirat analyzed the data, prepared figures and/or tables, authored or reviewed drafts of the article, and approved the final draft.
- Korakot Nganvongpanit conceived and designed the experiments, prepared figures and/or tables, authored or reviewed drafts of the article, and approved the final draft.
- Wanpitak Pongkan conceived and designed the experiments, prepared figures and/or tables, authored or reviewed drafts of the article, and approved the final draft.

## Animal Ethics

The following information was supplied relating to ethical approvals (*i.e.*, approving body and any reference numbers):

According to the Animals for Scientific Purposes Act, B.E. 2558 (2015), since a part of this experiment was performed on a Dugongs carcass from a Phuket Marine Biological Center, Phuket, Thailand, during the diagnosis procedure for the cause of death, no ethical approval was required for this study and confirmed by the Animal Ethics Committee, Faculty of Veterinary Medicine, Chiang Mai University.

## Data Availability

Raw data are available as a Supplemental File.

## Supplemental Information

Supplemental information for this article can be found online at http://dx.doi.org/10.7717/peerj.15859#supplemental-information.

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
