# Peer review of "Histological study of seventeen organs from dugong (Dugong dugon)"

_PeerJ, doi:10.7717/peerj.15859_

## Round 0.1 · original submission · Major Revisions

Dear Authors,

As per the recommendations of our expert reviewers, the manuscript attracts some issues to be addressed with justification and incorporations.

Therefore, I invite you for major revisions.

Please do the needful and resubmit asap.

Good luck

·

Basic reporting

See my detailed report below.

Experimental design

See my detailed report below.

Validity of the findings

See my detailed report below.

Additional comments

Thank you for giving me the opportunity to review this good piece of work. The authors have conducted a very extensive work and are suitable for publication in PeerJ, however, some points need to be addressed for the enhancement of the quality of the article. My points are mentioned below-
1. First of all English is not consistent with several grammatical errors. I suggest authors to rewrite the whole manuscript carefully and take the help of a fluent English writer.
2. In M & M specify how many samples were taken and the number of animals (males and female),
3. Line 76, what is the meaning of this sentence? Unfortunately, the histological information on the health and disease of dugongs is still lacking. Are you comparing healthy and diseased animals?
4. The discussion section should be continuous without subheadings, which surely attracts the readers.
5. Some of the figures are not clear and visualize the anatomical structure properly, replace or change the figures. The labeling in the figure is small and should be adjusted in the revisions.
6. Avoid using the old references from the manuscript and arrange all the references as per the standard form at of the journal PeerJ.

·

Basic reporting

The language and grammar requires revision in the whole manuscript. If possible kindly take help of online software

Introduction
Most humbly it is brought to your kind notice that the introduction was found to be vague and not following the standard structure. The following suggestions may be followed for rewriting the introduction:
• first the topic is introduced with standard but brief chronological review up to recent times,
• Based on introduction, then the problem/ lacuna in research is recognized and hypothesis is put forward.
• Then based on lacunae in the concerned field, objectives of the investigation are given.

Kindly review some recent research on the topic chronologically.’
Kindly include the well documented research for reduction in the dugong population. Such information wil increase the impact of research.

Figures are relevant, high quality, well labelled & described properly.

Raw data was supplied.

Experimental design

The presented research is original primary research within Scope of the journal.

Research question were well defined in the introduction
Material and methods are relevant & meaningful.

It is stated how the research fills an identified knowledge gap.

Rigorous investigation was performed to a high technical & ethical standard.

Comments for Material Methods-
Kindly mention the reference of the protocol of histology followed in the investigation.

Validity of the findings

The research will have good impact in concerned field, and it is a unique novel work.

Additional comments

Most humbly it is brought to your kind notice that the discussion started and ended abruptly. The following suggestions may be followed for a little modification in the discussion:
• First the topic is introduced by mentioning the hypothesis/reasons for research and objectives in brief with standard but brief chronological review up to recent times. For example “the presented study was conducted to investigate ………”. Followed by brief review “earlier research had showed …………”
• Discussion may end with remarks briefly summarizing the salient findings in logical sequence to draw a meaningful conclusion.

---

## Round 0.2 · accepted · Accept

Dear Authors,
With pleasure I inform that your manuscript entitled "Histological study of Seventeen organs from Dugong (Dugong dugon)" - has been Accepted for publication in PeerJ. This is an Academic acceptance and further needs certain publications tasks to complete, so I request you to keep you available for few days to avoid any delays. I wish you all the best for further submissions.
Good luck

·

Basic reporting

Good

Experimental design

Good

Validity of the findings

Good

Additional comments

Good

·

Basic reporting

No comments

Experimental design

No comments

Validity of the findings

No comments

Additional comments

No comments